# Facilitating Imaging Mass Spectrometry of Microbial Specialized Metabolites with METASPACE

**DOI:** 10.3390/metabo11080477

**Published:** 2021-07-23

**Authors:** Don D. Nguyen, Veronika Saharuka, Vitaly Kovalev, Lachlan Stuart, Massimo Del Prete, Kinga Lubowiecka, René De Mot, Vittorio Venturi, Theodore Alexandrov

**Affiliations:** 1Structural and Computational Biology Unit, European Molecular Biology Laboratory, 69117 Heidelberg, Germany; don.nguyen@embl.de (D.D.N.); veronika.saharuka@embl.de (V.S.); vitaly.kovalev@embl.de (V.K.); lachlan.stuart@embl.de (L.S.); 2Photolab, European Molecular Biology Laboratory, 69117 Heidelberg, Germany; massimo.del.prete@embl.de (M.D.P.); kinga.lubowiecka@embl.de (K.L.); 3Centre of Microbial and Plant Genetics, KU Leuven, 3000 Leuven, Belgium; rene.demot@kuleuven.be; 4International Centre for Genetic Engineering and Biotechnology (ICGEB), 34149 Trieste, Italy; Vittorio.Venturi@icgeb.org; 5Metabolomics Core Facility, European Molecular Biology Laboratory, 69117 Heidelberg, Germany; 6Molecular Medicine Partnership Unit (MMPU), European Molecular Biology Laboratory, 69117 Heidelberg, Germany; 7Skaggs School of Pharmacy and Pharmaceutical Sciences, University of California San Diego, La Jolla, CA 92093, USA

**Keywords:** imaging mass spectrometry, microbial natural products, spatial metabolomics

## Abstract

Metabolite annotation from imaging mass spectrometry (imaging MS) data is a difficult undertaking that is extremely resource intensive. Here, we adapted METASPACE, cloud software for imaging MS metabolite annotation and data interpretation, to quickly annotate microbial specialized metabolites from high-resolution and high-mass accuracy imaging MS data. Compared with manual ion image and MS1 annotation, METASPACE is faster and, with the appropriate database, more accurate. We applied it to data from microbial colonies grown on agar containing 10 diverse bacterial species and showed that METASPACE was able to annotate 53 ions corresponding to 32 different microbial metabolites. This demonstrates METASPACE to be a useful tool to annotate the chemistry and metabolic exchange factors found in microbial interactions, thereby elucidating the functions of these molecules.

## 1. Introduction

Imaging mass spectrometry (imaging MS) has become an invaluable tool for spatial metabolomics [1]. In microbiology and natural products research, imaging MS provides a capacity for in situ detection of specialized metabolites (i.e., small molecules synonymous with natural products, secondary metabolites, and also metabolites that serve functions specific to the producing microorganism) [2,3]. Finding MS ion images colocalized with visible phenotypes helps better pinpoint causal metabolites responsible for phenomena such as zones of inhibition or changes in microbial morphology [4,5]. As such, there is a growing interest in applications of imaging MS in natural products discovery, understanding the roles of specialized metabolites in microbial interactions, and characterizing metabolites involved in host–microbe or microbe–environment interactions [6,7].

One of the key challenges in imaging MS, metabolomics, and natural products discovery is metabolite identification, where discovery and characterization are often costly, time- and labor-intensive, and sometimes require large amounts of starting sample/material [4,8,9]. Specific challenges to imaging MS are large data sizes containing 10,000–1,000,000 spectra each corresponding to their own pixel, the lack of analyte separation prior to MS analysis, and the inability to perform data-dependent MS/MS in the imaging mode. To address these challenges for imaging MS in general, we recently developed a computational method for metabolite annotation in imaging MS data and implemented it as an open-source and free software called METASPACE [10,11]. METASPACE takes a high-mass accuracy high-resolution imaging MS dataset and performs MS1-based metabolite annotation against a user-selected metabolite database. METASPACE provides putatively annotated metabolites with their molecular formulas and corresponding ion adducts, metabolite ion images, values of the metabolite-signal match (MSM) score that ranks the annotations in a given dataset based on their annotation likelihood, and an estimated false discovery rate (FDR) value that quantifies the confidence of the metabolite annotation. Thus far, the databases on METASPACE revolve largely around lipids and small molecule mammalian metabolites [11,12]. Only recently have bacterial metabolite databases such as the *Pseudomonas aeruginosa* Metabolome Database (PAMDB) and the *E. coli* Metabolome Database (ECMDB) been introduced to METASPACE, and those are primarily centered around molecules in primary metabolism [13,14]. The Natural Products Atlas (NPA) is an open access and community-maintained microbial natural products database consisting of more than twenty-four thousand compounds connected to citations, structures, and source organisms [15,16]. Using METASPACE’s custom database creation feature, we created a new public database from NPA content and illustrate the interrogation of microbial agar imaging MS data from a group of actinobacteria, pseudomonads, and a *Bacillus*.

## 2. Results and Discussion

In total, we imaged 10 bacterial strains (Appendix A) consisting of 7 actinomycetes, 2 pseudomonads, and 1 *Bacillus* (Figure 1). These genera were selected because they collectively represent some of the most commonly associated microbes with specialized metabolite production [17,18,19]. These bacteria provide a wide range of potentially observable metabolites that can be found in NPA. For the strains in this study, NPA contains 5018 metabolites (Appendix A) when searching at the genus level and 250 metabolites (Appendix A) when searching at the species level. Broken down by genus, *Rhodococcus* has the smallest number of metabolites in NPA, with 20 entries, while *Pseudomonas* and *Bacillus* have 266 and 389 entries, respectively. *Streptomyces* has the largest number of entries at 4343 metabolites (Appendix A). When examining the number of metabolite entries in NPA at the species level, *Rhodococcus erythropolis* and *opacus* contribute 3 and 0 metabolites, respectively, to the total 20 entries in the genus *Rhodococcus*. *Pseudomonas putida* and *savastanoi* contribute 3 and 2 metabolites, respectively, to the genus *Pseudomonas*. *Bacillus subtilis* has a total of 101 entries out of the total 389. *Streptomyces griseus, coelicolor, lividans, mobaraensis*, and *roseolilacinus* each contribute 106, 25, 9, 1, and 0 entries, respectively, to the total of 4343 *Streptomyces* metabolites in NPA (Appendix A).

The 10 bacteria used here were imaged on the same plate in one acquisition to capture a snapshot of the specialized metabolites simultaneously produced (Figure 2 shows exemplary ion images with the full dataset available at https://metaspace2020.eu/annotations?db_id=36&ds=2020−01−07_11h15m21s (accessed on 20 July 2021), Appendix A). From this imaging MS dataset and using the NPA database, METASPACE annotated 18, 29, 95, and 1968 ions at the FDR levels of 5%, 10%, 20%, and 50%, respectively (Appendix A). Closer inspection revealed that some of those annotations represent false positives, which becomes evident based on either their molecular properties (e.g., some fungal metabolites were reported among annotations) or their ion images (e.g., images with intensities covering areas well outside of the colonies likely corresponding to MALDI matrix signals). We have considered two strategies for improving the results. The first strategy employs the off-sample filter implemented in METASPACE [20]. This filter uses an artificial intelligence approach to recognize images corresponding to off-sample areas, here associated with unspecific background, such as signals from MALDI matrix or agar. Due to ionization suppression, these signals would normally have lower intensities on and within the colony areas and can therefore be automatically determined. This filter found a total of 12, 20, 64, and 1045 annotations at the FDR levels of 5%, 10%, 20%, and 50% respectively (Appendix A). These numbers of annotations after off-sample filtration represent 67%, 69%, 67%, and 53% of all (both off-sample and on-sample) annotations and thus demonstrate its usefulness in the context of microbial colony analysis by METASPACE.

The second strategy we considered to reduce the false positive annotations was to filter metabolites associated with the particular cultured bacteria, based on taxonomic classification in NPA. This strategy includes (1) filtering out ions corresponding to background (area outside of the colonies), (2) selecting only annotations matching on NPA with genera of our cultured bacteria (*Bacillus, Pseudomonas, Streptomyces,* and *Rhodococcus*), and (3) selecting only annotations matching on NPA with the species of our cultured bacteria. The species filter proved to be overly stringent because this metadata field is often unpopulated in NPA (Appendix A). In addition, we believe that if an annotated molecule has not been previously detected in a given bacterial strain, but this molecule is also observed in a related and already characterized strain, the annotation is likely correct (although traditional metabolite identification or dereplication workflows would be required for confirmation).

After filtration, we manually curated the annotations so that each ion image exhibits a localization within a colony or near to the colony assuming possible secretion. Moreover, we selected annotations corresponding to one bacterium only (Figure 2 and Appendix A). In total, this process resulted in 53 curated annotations: *R. erythropolis*, *R. opacus*, and *S. roseolilacinus* all yielded zero annotations, *P. savastanoi* and *S. lividans* each had one annotation, *S. griseus* and *S. coelicolor* had three annotations, and *B. subtilis* and *P. putida* both had nine annotations (Appendix A). These data show the effects of considering higher quality annotations (at FDR 20% vs FDR 50%) or applying off-sample on the genus- and species-level annotations. For example, for the genus *Bacillus**,* after off-sample filtering, 26 annotations were left, which represents 65% of all annotations at FDR 50% and is consistent with the overall effect of the off-sample filter leaving 67% (median value) of annotations. Examining high-quality annotations at FDR 20% has a more dramatic effect, leaving only two annotations for the genus *Bacillus*. These trends are also observed for the remaining organisms used in this study (Appendix A). Interestingly, *S. mobaraensis* had the largest number of curated annotations at 27 in total (Appendix A). At the genus level, the numbers of annotations in this study reflect the trend of an increasing number of entries within NPA: *Rhodococcus*, *Pseudomonas*, *Bacillus*, and *Streptomyces* each have, respectively, 20, 266, 389, and 4343 entries versus 4, 39, 40, 512 metabolites annotated, respectively (Appendix A). However, the same trend was not followed at the species level. *S. mobaraensis* has the greatest number of curated annotations at 27, while *S. griseus* and *S. coelicolor* each have only 3 annotations. Within NPA, the *Streptomyces* species *mobaraensis*, *coelicolor*, and *griseus* have 1, 25, and 106 entries, respectively. We believe there are several factors that can explain this observation. Generally speaking, there are three cases why we do not observe metabolite annotations: (1) non-conducive biological conditions, (2) technical MS nuances, and (3) METASPACE limitations.

(1)Lack of metabolite annotation due to inappropriate culture conditions. For example, the growth conditions used here may not be conducive to specific metabolite production [21]. In certain cases co-cultivation of multiple organisms may be required for metabolite elicitation [21,22]. Conversely, the presence of multiple organisms growing in close proximity may alter or even abolish metabolite production and therefore lead to lack of metabolite annotation by METASPACE. Or simply the microbes were not incubated for the correct duration to observe specific metabolite production. Insufficient culturing time may lead to metabolite analogs or incomplete metabolites altogether.(2)Lack of metabolite annotation due to technical MS aspects. Most importantly are sample preparation steps, specifically matrix application. Different matrices increase or decrease metabolite ionization. In order to maximize the number of metabolites observed, one would have to prepare multiple samples with multiple MALDI matrices and acquire data in both MS polarities. Even then, there could be metabolites that are produced by the microbes but these metabolites could be below instrument detection limits. In addition, potentially annotatable molecules that are present within NPA may not be observed if these molecules fall outside of the user defined MS settings. Less likely, but still theoretically plausible, are the events of uncommon adduct formation, which would not be taken into consideration by METASPACE. At the moment, METASPACE takes into account the most common MS adducts observed: +H, +Na, and +K, but METASPACE does have the ability to generate other adducts from chemical formulas such as [M]+ adducts, M+metal adducts, or M+adduct-neutral loss.(3)Lack of metabolite annotation due to METASPACE. The most important criteria for metabolite annotation is appropriate database selection in METASPACE. For example, if experimental data containing microbial samples were annotated against the more common databases on METASPACE, such as the Human Metabolome Database (HMDB), then the majority of the annotations would be false positives. Likewise if experimental data containing mammalian cell culture samples were annotated against microbial databases, such as NPA or PAMDB, the annotations here would similarly consist of false positives. Furthermore, database curation issues may also contribute to lack of annotation, for instance, if a bacteria is a known producer of a metabolite, but this entry has not been added to the database, then this annotation will be overlooked until the database is updated. Finally, if one were so inclined, a large database such as NPA could be filtered so that only metabolite entries of a single organism would be curated and then uploaded to METASPACE. This would be an extreme example of targeted imaging MS analysis.

Coming back to the total number of annotations in this dataset, we also asked ourselves why there are so many annotations, 1968, as compared to annotations at FDR 20% with off-sample filtering, 64, and to annotations we believe to be correct, 53 (Appendix A). We believe that the large number of total annotations can be attributed to the high number of background signals from agar (Appendix A, and Appendix A). As we suspected, the number of annotations at FDR 50% dropped for each genus after off-sample filtering. *Bacillus*, *Pseudomonas*, *Rhodococcus*, and *Streptomyces* annotations drop from 40 to 26, 39 to 27, 4 to 1, and 512 to 285, respectively after off-sample filtering (Appendix A). The decreasing number of annotations with and without off-sample filtering is not as drastic at FDR 20% because there are less annotations to begin with due to higher annotation requirements. *Bacillus* annotations remain at 2 with and without off-sample filtering, *Pseudomonas* annotations drop from 2 to 1 with and without off-sample filtering, *Rhodococcus* does not have any annotations at FDR 20%, and *Streptomyces* annotations drops from 30 to 22 with off-sample filtering (Appendix A). We recommend that high FDR levels (50%) should be used with caution and may require manual curation and inspection due to an overall low quality of signals and high risk of false positive annotations.

In principle, samples with high background signals, as is usually the case with agar-based MALDI imaging MS, have *m*/*z* values that will coincidentally match entries in the database. Coincidental matching can occur no matter the sample type or database. The large number of background signals matching to decoy formulas can unfortunately mix with correct annotations, thereby obscuring potentially valuable data. However, we believe that METASPACE bypasses these issues for two reasons. Off-sample filtering represents a viable strategy for removing a large part (33% on average) of annotations corresponding to background signals. Additionally, specialized metabolite production is so tightly linked to its taxonomic origin that we can leverage information such as source origin and species to quickly filter a large number of annotations.

## 3. Materials and Methods

### 3.1. Microbial Cultures of Actinomycets, Pseudomonads, and Bacillus

All microbes were grown in NB liquid (5 g peptone, 3 g meat extract, and 1 L Milli-Q H_2_O) and NB agar (5 g peptone, 3 g meat extract, 15 g agar, and 1 L Milli-Q H_2_O). Petri dishes used for NB agar were 80 mm in diameter and filled with 10 mL of agar. Because these bacteria have drastically different growth rates, inoculation into NB liquid and then onto NB agar was staggered so that colonies would grow to approximately the same size on the day of MALDI imaging. Frozen stocks of each bacteria were thawed, and then 20 μL of the stock was inoculated into 7 mL of NB liquid and incubated at 30 °C and 150 RPM. Next, 1 μL of the liquid culture was then inoculated onto an NB agar plate in the locations specified in Figure 2. Plates were parafilmed and incubated on the bench at room temperature. The exact timeline of the liquid culture and agar inoculations is detailed in the table below.
Time (hours)04872ActionInoculate and incubate liquid cultures for:*S. griseus**S. coelicolor**S. lividans**S. roseolilacinus*Inoculate and incubate agar cultures for:*S. griseus**S. coelicolor**S. lividans**S. roseolilacinus*Inoculate and incubate liquid cultures for: *P. savastanoi**R. erythropolis**R. opacus**S. mobaraensis*Time (hours)120144168ActionInoculate and incubate liquid cultures for:*B. subtilis**P. putida*Inoculate and incubate agar cultures for:*P. savastanoi**R. erythropolis**R. opacus**S. mobaraensis*Inoculate and incubate agar cultures for:*B. subtilis**P. putida*Photograph samples, cut out samples and transfer to glass slide, dry sample overnight

### 3.2. Imaging MS of Microbial Cultures

Optical images of the colonies were taken using digital full frame Canon cameras with a prime 100 mm f/2.8 macro lens. The camera was mounted at 90 degrees above the subject, and one flash light from the side was used for scene lighting. The camera settings were f/4.5, shutter speed 1/125th of second, and 100 ISO, and a ruler was added to the photos and used as a scale. Colonies and the surrounding agar were cut and removed from the petri plate and transferred to a standard 75 × 25 mm glass microscope slide. Samples were dried uncovered overnight on the bench, at which point the sample was flush with the glass slide. Samples were coated with MALDI matrix (1:1 mixture by mass of 2-5-dihydroxybenzoic acid (2,5-DHB, Sigma-Aldrich) and α-cyano-4-hydroxy-cinnamic acid (CHCA, Sigma-Aldrich), at a concentration of 20 mg/mL in 70% ACN (LC/MS grade, Optima, Fisher Scientific): 30% H_2_O (Milli-Q);) using an HTX TM-Sprayer (HTX Technologies, LLC, Chapel Hill, NC, USA) with the following spray settings: spray temperature: 80 °C; passes: 6; flow rate: 0.133 mL/min; velocity: 1000 mm/min; track spacing: 2 mm; pattern: CC; pressure: 10 psi; gas flow rate: 5 mL/min; drying time: 2 s; and nozzle height: 40 mm. MALDI imaging MS was performed using an AP-SMALDI5 AF source (TransMIT GmbH, Giessen, Germany) with 20% laser filter on, attenuator at 30°, in 3D mode for laser autofocus at each pixel, 45 micron step size, and over an area of 400 × 900 pixels. Data was acquired on an orbital trapping mass analyzer, QExactive Plus MS (Thermo Fisher Scientific Inc., Bremen, Germany) using MS full scan between 106.7–1600 *m*/*z*, at a resolution of 140,000, in positive mode, at 1 microscan, with an AGC target of 1e6, and a maximum inject time of 500 ms. Spray voltage was kept at 3.00 kV, with a capillary temperature of 250 °C, and a S-lens RF level of 50.0.

### 3.3. Database Curation and Molecular Annotation

Thermo RAW files were converted to mzMLwith MSConvert using the following settings: binary encoding precision at 64-bit, write index, use zlib compression, TPP compatibility checked, and with peak picking filter turned on. The resulting mzML file was then converted to imzML using imzML Converter version 1.3 [23] with the following settings: pixels in x: 400; pixels in y: 900; file organization: image per file; storage type: processed; *m*/*z* array data type: 64-bit float; intensity array data type: 32-bit float; line scan direction: linescan left right; scan direction: top down; scan pattern: flyback; and scan type: horizontal line scan. The resulting .imzML and .ibd files were uploaded to METASPACE for annotation and can be downloaded and viewed here: https://metaspace2020.eu/api_auth/review?prj=4e85e354-af2b-11ea-b3af-8f79dfd1b160&token=XAvi9QAeWyQY (accessed on 20 July 2021). To generate an average mass spectrum of the entire imaging surface (Appendix A), the imzML was loaded into MSiReader (v1.01) [24,25] and the polygon tool was used to select the region of interest. Upon doing so, the average mass spectrum of the selected pixels can then be exported and viewed. Molecular annotation by METASPACE was performed using the database information from the Natural Products Atlas (NPA, version 2019-08) [16]. A custom database was created in METASPACE by going to the user’s account, groups, and databases. Upon clicking “Upload Database”, database name and version number were filled out followed by an upload of the formatted database file. Database information from NPA were formatted for METASPACE compatibility with the column headers as follows: annotation identification number, molecule name, molecular formula, and if present (but not required) InChi keys for molecular visualization. Extra information such as publication reference, molecular biological source, and source taxonomic information can be included but is also linked to NPA directly via METASPACE.

Off-sample filtering is a feature built directly into METAPSACE. To show/hide off-sample filtering, from the dataset, click “Add filter” and select “Show/hide off-sample annotations”. The “Diagnostics” tab shows whether the current annotation is on-sample or off-sample. For curating annotations with taxonomic information, the NPA database version 2019-08 was downloaded from the NPA. For each METASPACE annotation, all corresponding NPA IDs were obtained, and metadata from the field’s source organism type/origin type (all, bacterium, or fungi), source genus, and source species were collected. Annotations where source organism type/origin type did not match to “Bacterium” were eliminated. For genus-level filtering, annotations where genus matched to “*Bacillus*”, “*Pseudomonas*”, “*Rhodococcus*”, or “*Streptomyces*” were retained. The script for filtering METASPACE annotations can be found at https://github.com/alexandrovteam/microbial-metaspace (accessed on 20 July 2021).

## 4. Conclusions

We demonstrated that METASPACE, in combination with an appropriate specialized metabolite database, has the capability to annotate microbial specialized metabolites from agar-based imaging MS data. METASPACE was able to annotate 53 ions corresponding to 32 specialized metabolites that we verified to be correct based on a match to the correct taxonomic classification contained within NPA. We would like to note that annotations of these metabolites are putatively correct based on MS1 properties (exact *m*/*z*, isotopic pattern, measure of spatial chaos, co-localization of isotopic ions). To be fully confident, additional information, such as MS/MS experiments or orthogonal techniques, such as molecule purification and subsequent NMR structure elucidation, may be required. Depending on the questions being asked, an MS1 level database match in combination with correct biological information (i.e., the correct organism, genus, and potentially species, are producing feasible molecules) and appropriate metabolite localization within the ion image can provide valuable information for hypothesis generation and screening as well as for quality control or optimization culturing conditions. In instances where we need to determine what metabolites are present in imaging MS data, METASPACE is a powerful tool and provides capacities for visualization, sharing, and publication of microbial imaging MS data and annotations. In the future, we believe that METASPACE can be applied towards studying metabolic exchange to examine the causal metabolites responsible for an observed phenotype, as well as guide the discovery of novel specialized metabolites.

## Figures and Tables

**Figure 1 metabolites-11-00477-f001:**
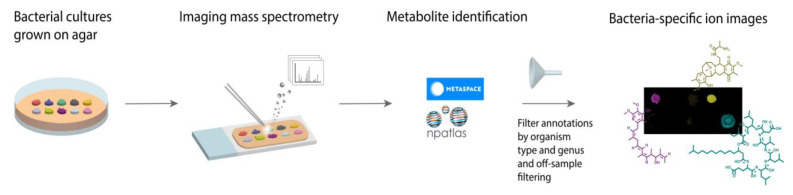
Workflow for microbial imaging mass spectrometry (imaging MS) followed specialized metabolite annotation using METASPACE. Bacterial colonies were grown on agar plates, coated with MALDI matrix, and subjected to imaging MS. The resulting data was submitted to METASPACE with the Natural Products Atlas (NPA) selected as a search database. The subsequent annotations were filtered by organism type and genera or off-sample filtering.

**Figure 2 metabolites-11-00477-f002:**
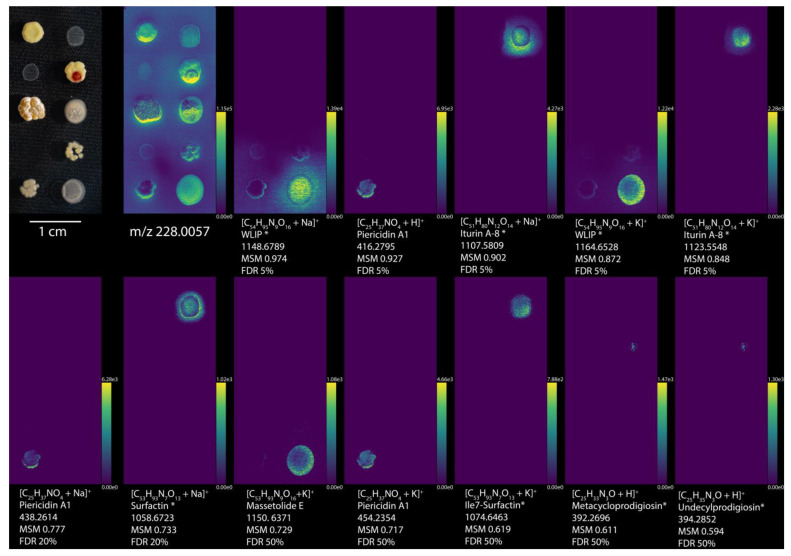
Imaging MS and curated molecules by METASPACE using NPA. All annotations were filtered by organism type, genus, and then hand selected for reasonable ion distribution, resulting in the 12 highest ranked annotations. Optical images of *Streptomyces griseus* subsp. *griseus* 40236, *Bacillus subtilis* BGA 618, *Pseudomonas savastanoi* 722, *Streptomyces coelicolor* A2(3)/M145, *Streptomyces lividans* PM66, *Rhodococcus erythropolis* 43,066, *Rhodococcus opacus* 43,205, *Streptomyces roseolilacinus* 40,173, *Streptomyces mobaraensis*, and *Pseudomonas putida* RW10S2, from left to right, top to bottom. Ion images are arranged from increasing to decreasing metabolite-signal match (MSM) score from left to right, top to bottom, while *m*/*z* 228 was selected as a background ion image to show the locations of all the bacteria. Each ion image is denoted by the ion detected, molecule name from NPA, the monoisotopic *m*/*z* value, MSM score, and false discovery rate (FDR) percentage. Molecule names with an * represent annotations with potential isomers. Ion images are available at https://metaspace2020.eu/annotations?db_id=36&ds=2020-01-07_11h15m21s&fdr=0.5&viewId=goxJCwqI&hideopt=1 (accessed on 20 July 2021).

## Data Availability

The data presented in this study is available on METASPACE: https://metaspace2020.eu/project/nguyen-2021-microbial_metaspace, (accessed on 20 July 2021).

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
