# Peer review of "Facilitating Imaging Mass Spectrometry of Microbial Specialized Metabolites with METASPACE"

_metabolites, 2021, doi:10.3390/metabo11080477_

Round 1
Reviewer 1 Report
I have enjoyed reading your paper and I am always very glad to see scientific data being easily accessible to the reviewers and readers. Thanks for providing all the metaspace links. I have several comments and suggestions with regards to the contents of the paper.
- While I expect the mass spectra of the bacterial colonies not to be too pretty, I think any paper that tries to publish on imaging mass spectrometry should include at least one mass spectrum in the main article or in the SI. Without seeing what the mass spectra look like I think it’s difficult to have an idea what the annotation pipeline (in this case metaspace) is actually annotating. Could the authors include either:
- Mean spectrum of the entire dataset – probably going to be dominated by the background since the colonies occupy minority of the slide.
- Mean spectrum of the entire dataset without background.
- Mean spectrum of each of the colonies – this we can see how different the colonies are.
- The main point of your paper is to promote metaspace as a tool for bacterial metabolite annotation, however, it would seem to me that the number of putative annotations is far too high to be reasonable for MS1 level of annotations. Of course you rightly point out that number of annotations is roughly proportional to the number entries for particular culture in the underlying database. My concern is that the final 53 annotations required a lot of curation to remove the false positives. And even after that, a lot of the ions in the list of 53 have extremely low MSM scores at high FDR which would also indicate that they are false positives. Would you mind commenting on this and add more text to the manuscript that makes it clearer that many of the selected annotations would be filtered out based on the metaspace metrics. Also, I think it would be a good idea to add the MSM and FDR metrics to the Figure 2, Figure S1 and the table.
- Page 4 line 135-152. You list a number of reasons why certain colonies have not yielded any annotations or fewer than expected. Have you performed any of these experiments to confirm your suspicions? Have your performed some of these experiments, such as growing the individual colonies and then using metaspace to annotate them individually to check whether you obtain more/less/same level of annotations?
- Page 4-5 line 135-165. I find this paragraph to be try to say a lot but in a very disjointed manner by using many short sentences that seem that they’ve been finished prematurely. Perhaps the authors could consider rewriting this section so it reads more naturally.
Author Response
Please see attachment for reviewer 1 response.
Note: original uploaded file was titled "Reply_to_reviewer_1.docx" and has been changed to "auothor-coverletter-12602542.v1.docx" upon upload to the portal.

Reviewer 2 Report
This is a nice paper on ion annotation in imaging MS.
Does one have the ability to upload an in-house library/database into METASPACE to be used for ion annotation?
Author Response
Please see attachment for response to reviewer 2.
Note: original uploaded file was titled "Reply_to_reviewer_2.docx" and has been changed to "auothor-coverletter-12608195.v1.docx" upon upload to the portal.

Round 2
Reviewer 1 Report
Dear Authors,
Thank you for making the requested changes and clarifying the text. I think it improved the manuscript and I am now happy to approve the changes.
Thermo’s QualBrowser could only average ~25 minutes worth of data (~1500 scans), while the entire dataset is 5905.44 minutes long (360,000 scans in total). In this case, the average mass spectrum took over 30 hours to generate.
All I can say is that this is shocking and should not be acceptable…
I hope that you are aligned with our reasons and agree with us that to describe such data would be outside the scope of this short manuscript.
Thanks for clearing this up and indeed, I am satisfied that it would outside of the scope for this manuscript.